# The ULR-repro3 GPS data reanalysis and its estimates of vertical land motion at tide gauges for sea level science

*Médéric Gravelle[1], Guy Wöppelmann[1], Kevin Gobron[1,2],*

*Zuheir Altamimi[3,4], Mikaël Guichard[1], Thomas Herring[5], Paul Rebischung[3,4]*

[1] LIENSs, CNRS – La Rochelle University, 17000 La Rochelle, France
[2] Royal Observatory of Belgium, Uccle, Belgium
[3] Université de Paris, Institut de physique du globe de Paris, CNRS, IGN, F-75005 Paris, France
[4] ENSG-Géomatique, IGN, F-77455 Marne-la-Vallée, France
[5] MIT, Cambridge, MA, USA

*Correspondence to*: Médéric Gravelle (mederic.gravelle@univ-lr.fr)

## Abstract

A new reanalysis of Global Navigation Satellite System (GNSS) data at or near tide gauges worldwide was produced by the University of La Rochelle (ULR) group within the 3rd International GNSS Service (IGS) reprocessing campaign (repro3). The new solution, called ULR-repro3, complies with the IGS standards adopted for repro3, implementing advances in data modelling and corrections since the previous reanalysis campaign, and extending the average record length by about 7 years. The results presented here focus on the main products of interest for sea level science, that is, the station position time series and associated velocities on the vertical component at tide gauges. These products are useful to estimate accurate vertical land motion at the coast and supplement data from satellite altimetry or tide gauges for an improved understanding of sea level changes and their impacts along coastal areas. To provide realistic velocity uncertainty estimates, the noise content in the position time series was investigated considering the impact of non-tidal atmospheric loading. Overall, the ULR-repro3 position time series show reduced white noise and power-law amplitudes and station velocity uncertainties compared to the previous reanalysis. The products are available via SONEL (https://doi.org/10.26166/sonel_ulr7a; Gravelle et al., 2022).

# 1 Introduction

Vertical land motion plays a crucial role in understanding sea level changes and its spatial variability (Wöppelmann & Marcos, 2016; Frederikse *et al*., 2020; Hamligton *et al*., 2020; for recent reviews and references therein). This is especially true along the coasts, where its monitoring is often an essential requirement to assess the extent of the environmental and socio-economic threats posed by changing sea levels in a warming climate at regional or local scales (Magnan *et al*., 2020). Changes in sea level can be measured relative to the land by tide gauges, or relative to the Earth's centre of mass by satellite altimeters (e.g., Marcos *et al*., 2019). In both relative (tide gauge) and geocentric (satellite) measuring systems, accurate estimates of vertical land motion are essential, either to disentangle the solid Earth contribution from other factors in tide gauge records (Woodworth *et al*., 2019), or to supplement satellite altimetry data to assess relative sea level change for coastal studies and planning (Poitevin *et al*., 2019).

In the last decades, significant efforts have been undertaken to produce accurate estimates of vertical land motion at tide gauges using Global Navigation Satellite System (GNSS) data (e.g., Sanli and Blewitt, 2001; Wöppelmann *et al*., 2007; Hammond *et al*., 2021). Wöppelmann *et al*. (2007) showed the importance of applying a homogeneous GNSS data reanalysis strategy across the entire data span, that is, using the same modelling, corrections and parameterization, to address the demand of accurate position time series and velocities for sea level studies. This conclusion was reached independently by Steigenberger *et al*. (2006) within the International GNSS Service (IGS; Johnston *et al*., 2017). Since then, the IGS has conducted several data reanalysis campaigns, stimulated by progress in modelling and corrections, lengthening of measurement records and updates of the International Terrestrial Reference Frame (ITRF) realizations (Rebischung *et al*., 2016).

In 2019, the IGS launched a third reprocessing campaign, designated as 'repro3', involving the international GNSS community (Rebischung, 2021). The University of La Rochelle (ULR) group contributed to this effort with a solution (ULR-repro3), which specifically includes a large selection of reliable GNSS stations near tide gauges. This paper describes the latest ULR solution in a series, succeeding previous releases described in Wöppelmann *et al*. (2009) and Santamaria-Gomez *et al*. (2017). This solution complies with the modelling and corrections adopted for 'repro3' (Rebischung, 2021; http://acc.igs.org/repro3/repro3.html), for example, corrections are made for antenna phase centre and solid Earth tides (see Section 2.2.1) . It

specifically highlights the time series of station positions and their vertical velocities, which are
the main products of interest for the sea level community. A crucial piece of information for
the practical use of these products is their uncertainties, which must account for the presence of
time-correlated stochastic variations (or noise) in the position time series (Williams *et al.*,
2004). Consequently, this paper also presents the statistical modelling strategies employed to
derive realistic uncertainty estimates. These results are presented together with a comparison
with respect to the previous ULR solution to appraise the progress accomplished over the past
seven years.

## 2 The ULR-repro3 products

### 2.1 Input data

Although the term GNSS is employed throughout the manuscript, the ULR-repro3 reanalysis
considered Global Positioning System (GPS) observations only. The GNSS measurements were
retrieved from the SONEL archive (www.sonel.org) in the form of station-specific daily files
in the international standard RINEX format (https://igs.org/wg/rinex/). These contain dual-
frequency carrier phase and pseudo-range measurements with a typical sampling of 30s.
SONEL holdings include data from over 1,200 stations around the world, amounting to over
6,300,000 daily files. A station selection was applied with the criteria of targeting time series
with over three years of continuous GNSS measurements and 70% completeness, located at or
near a tide gauge (within 15km). The term "continuous" denotes that no offset discontinuity in
the station position was anticipated from the metadata available, that is, from the station
operation logfiles, which should report changes in instrumentation, or from the co-seismic
displacements predicted using the earthquakes database and modelling described by Métivier
*et al.* (2014), updated to 2020. Some exceptions to these selection criteria concerned the French
GNSS stations at tide gauges, as part of the ULR commitment for France to the global sea level
observing (GLOSS) programme of the Intergovernmental Oceanographic Commission. This
programme was initiated in 1985 to establish a well-designed, high-quality *in situ* sea level
observing network to support a broad research and operational user base. Its primary products
are sea levels from permanent tide gauges provided with different sampling rates, data latencies
and averaging periods (IOC, 2012). SONEL is one of the five global data centres of GLOSS,
dedicated to assembling raw measurements from permanent GNSS stations at or near tide
gauges, as well as products of their analysis (GNSS position time series and velocities).

The spatial distribution of the GNSS stations considered in ULR-repro3 is shown in Figure 1 with symbols coloured according to the record length, ranging from 3 months to 21 years. The last year processed is 2020, instead of 2013 for the previous ULR reanalysis (Santamaria-Gomez et al., 2017), reaching an overall extension of seven years with a median station record length of 13.1 years. The station network shows a global distribution (Figure 1) with stations that are obviously far from coastlines: they were added from the IGS repro3 station priority list as reference frame stations to ensure an optimal alignment to the International Terrestrial Reference Frame (ITRF) and estimation of the satellite orbits. The ULR-repro3 station network ultimately consists of 601 GNSS stations (Figure 1), among which 176 are reference stations.

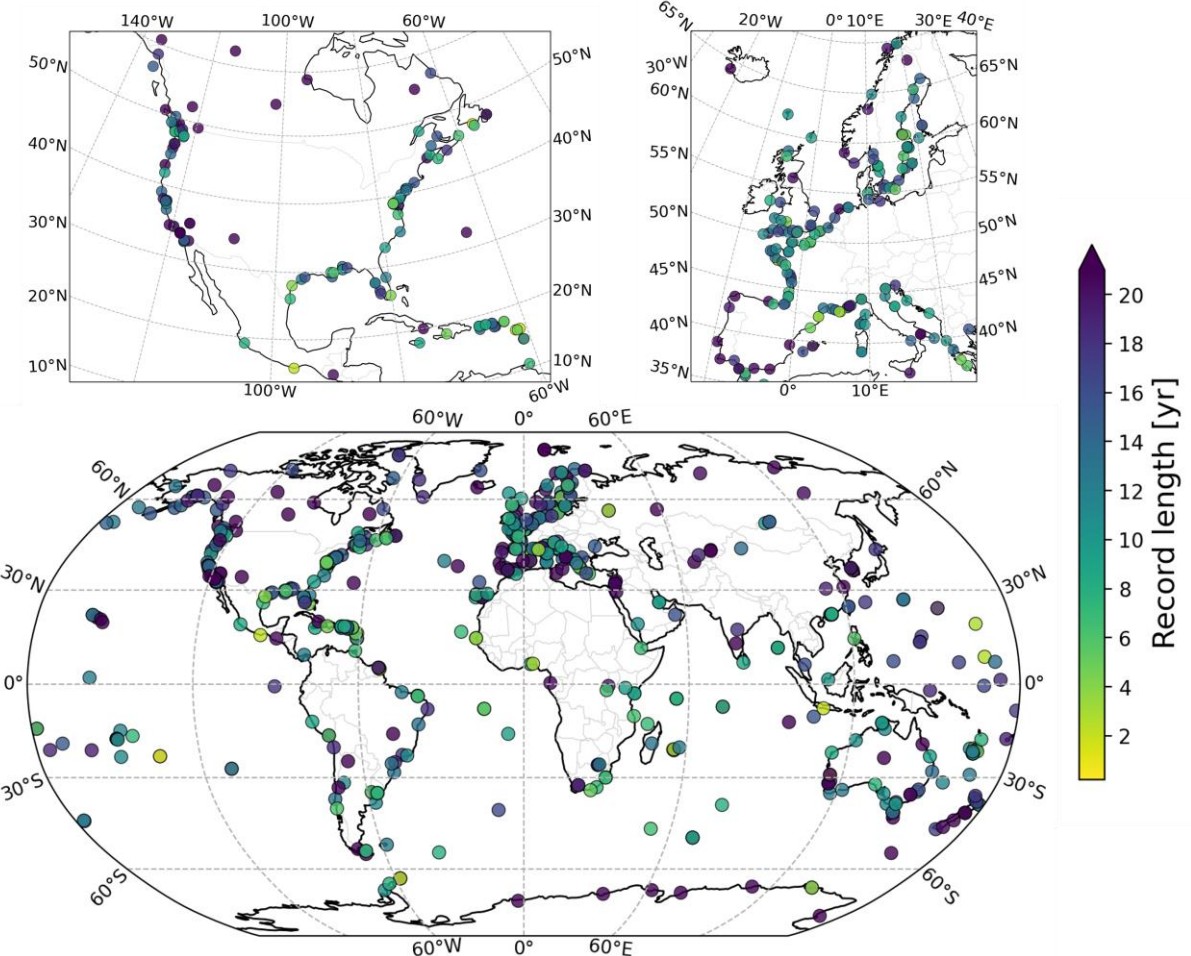

**Figure 1**: Spatial distribution of the 601 GNSS stations in ULR-repro3 and record length (colour bar), whose median is 13.1 years, spanning the 2000.0-2021.0 period.

## 2.2 GNSS processing

Estimating accurate vertical land motion from GNSS measurements involves several essential steps, such as computing daily station positions or deriving trends from the position time series.

In the first step, many corrections are applied, and other parameters such as satellite orbits or
atmospheric delays are adjusted along with the station positions (details in Section 2.2.1). It
requires advanced modelling and corrections and is usually best performed in a free-network
approach or loosely constrained strategy (Heflin *et al.*, 1992; Altamimi *et al.*, 2002), whose
major output is a global set of daily station positions expressed in an undetermined terrestrial
frame. The next step is to align these global solutions of daily station positions to a stable and
well-defined terrestrial frame such as the ITRF2014 (Altamimi *et al.*, 2017). The last step
involves modelling the kinematics described by the position time series, to obtain the quantity
of interest (trends, periodic oscillations, step discontinuities, etc.). Each step involves analyst
choices that can affect the estimated quantity of interest, and subsequently the geophysical
interpretation. The details below can thus be crucial to understand the results and their
uncertainties.

*2.2.1 Modelling & Corrections*
The ULR-repro3 processing considered the advances that occurred over the past seven years,
since the second IGS reanalysis campaign (Rebischung *et al.*, 2016). It complies with the
highest international standards, which were adopted by the IGS for the third reprocessing
campaign (http://acc.igs.org/repro3/repro3.html). The new modelling and corrections were
implemented in the GAMIT/GLOBK software packages (Herring et al., 2015; 2018) used here,
in particular the International Earth Rotation and Reference Systems Service (IERS) linear pole
model adopted in 2018 and the high-frequency (subdaily) Earth Orientation Parameters (EOP)
tide model from Desai and Sibois (2016). Table 1 provides a summary of the main modelling
features and corrections applied in the ULR-repro3 reanalysis.


| ULR-repro3 modelling and corrections | |
|---|---|
| Observations | double-differenced phase observations (GPS only, L1 & L2) |
| Sessions and sampling | 24-hr sessions; 2 min. sampling (30s in the data cleaning) |
| Elevation cut-off angle | 10 degrees |
| Antenna phase centre | igsR3_2135.atx (IGS Mail by A. Villiger, Dec. 2020) |
| Ionosphere refraction | ionosphere-free linear combination (1st order effect); 2nd and 3rd order corrections using IGRF13 (Alken *et al.*, 2021) and IGS combined IONEX files. |
| Troposphere refraction | A priori zenith delays from Saastamoinen model, mapped with VMF1 functions (Böhm *et al.*, 2006); zenith wet delays estimated at 1-hr intervals and gradients in north-south and east-west directions at 24-hr intervals. |
| Gravity field model | EGM2008 up to degree and order 12 (Pavlis *et al.*, 2012) |
| Solid Earth tides | IERS conventions (Petit & Luzum, 2010) |
| Ocean tide model | FES2014b (Lyard *et al.*, 2021) |
| Mean pole | linear mean pole as adopted by IERS in 2018 |
| Subdaily EOP model | Earth Orientation Parameters tide model from Desai and Sibois (2016) |
| Ocean tide loading | Provided by the EOST loading service (J.P. Boy; http://loading.u-strasbg.fr) using the ocean tide model FES2014b (Lyard *et al.*, 2021) |

**Table 1**: Main features of the GNSS data analysis strategy adopted for ULR-repro3 following
the IGS recommendations (http://acc.igs.org/repro3/repro3.html).

The remaining aspects of the ULR-repro3 data analysis strategy align with the approach used
in Santamaria-Gomez *et al.* (2017), so the following only briefly outlines them to understand
the analyst choices for geophysical application and interpretation. For each network of stations,
double-differenced GPS phase observations were processed in the ionosphere-free L1/L2 linear
combination. To minimise the impact of mismodeled low-elevation tropospheric delays,
satellite observations below 10 degrees were not considered. This cut-off angle aims to mitigate
the limitation due to ground antennas without absolute calibration (13% of the antennas in the
ULR-repro3 network). These antennas have a relative calibration (with respect to an antenna
with absolute calibration) converted to absolute considering only elevation-dependent phase
centre variation (PCV) down to 10 degrees. For the other (calibrated) GNSS antennas, phase
centre offsets with azimuth-dependent and elevation-dependent absolute PCV corrections were
applied (igsR3_2135.atx; IGS Mail by A. Villiger, 2020). Satellite-specific antenna phase
centre offsets and block-specific nadir angle-dependent absolute PCV were applied for the
transmitting antennas.

The first-order ionospheric delays were removed using the ionosphere-free linear combination
observations, whereas the second and third orders were corrected using the International
Geomagnetic Reference Field model (Alken *et al.*, 2021) and total electron content maps from
the IGS IONEX files. For the tropospheric delays, a priori hydrostatic zenith delays at the
ellipsoidal surface were obtained for each station from the VMF1 grids (Böhm *et al.*, 2006).
They were then reduced to the station heights using the GPT2 model (Lagler *et al.*, 2013). The
residual zenith tropospheric delays were adjusted at 1-hr intervals (i.e., 25 parameters per day)
for every station using a piecewise linear model, assuming the unmodeled wet component
dominates. Both the hydrostatic and wet zenith tropospheric delays were mapped to the
observation elevations using the VMF1 functions. The azimuthal asymmetry in the tropospheric
delay was accounted for by estimating a linear change in gradients (north-south and east-west)
over each day and station using the mapping function from Chen and Herring (1997).

The phase observations were weighted by elevation angle in the first iteration, and then by
elevation angle and station-dependent scatter of the phase residuals obtained from the first
iteration. The double-differenced phase ambiguities were adjusted to real values except when
they could be confidently fixed to integer values (more than 85% fixed). Within the same
inversion, GNSS satellite orbital parameters were adjusted using 24-hr arcs, IGS orbits as a
priori values and loose constraints consistent with the station position constraints (free-network
approach). Non-gravitational constant and once-per-revolution accelerations on the satellites
were adjusted too, using the ECOMC model. This model is a combination of the ECOM1 and
ECOM2 models (Springer *et al.*, 1999; Arnold *et al.*, 2015) with specific parameters
constrained in post-processing. Nominal satellite attitude corrections were applied, except
during eclipse periods where yaw rates were modelled (Kouba, 2009). Phase rotations due to
changes in the satellite antenna orientation away from the Earth-pointing direction were also
applied (Wu *et al.*, 1993). Regarding the Earth orientation parameters (pole position, rate and
length of day), these were estimated daily with a priori values from the IERS Bulletin A.
Modelled diurnal and semi-diurnal terms were added to the a priori pole and UT1 values
following the IERS Conventions (Petit and Luzum, 2010).

Note that neither loading displacements due to atmospheric tides nor non-tidal (atmospheric,
oceanic, hydrologic) loading displacements were corrected during the first step, which aimed
at estimating daily station positions from the GNSS measurements. By contrast, the
displacements of the crust due to solid-Earth and pole tides (solid Earth and ocean) were
corrected following the IERS Conventions (Petit and Luzum, 2010). Crustal motion due to the
ocean tide loading was corrected too, using the tidal constituents computed by EOST loading
service at each station from the FES2014b model (Lyard *et al*., 2021).

### 2.2.2. Offset detection & terrestrial frame alignment

Figure 2 shows the number of stations selected for ULR-repro3 with GNSS observations
available each day over the time period considered (2000.0-2021.0), ranging from 110+ to
nearly 500 stations. For computational efficiency, the stations were split into several (up to 10)
regional subnetworks, each having between 29 and 70 stations processed independently. For
the reader interested in this technicality, Figure S1 shows the regional subnetworks distribution
for the day 2018-01-01. An additional subnetwork of globally distributed stations was
considered to allow the daily combination of the regional subnetwork results in a unique daily
global solution. This global subnetwork was made up of IGS reference frame stations, each of
which also appeared in one – and only one – of the regional subnetworks. In turn, one regional
subnetwork included one IGS reference frame station at least, but could include more depending
on the total number of subnetworks. Moreover, to strengthen the physical link between regional
subnetworks, two stations from adjacent regional subnetworks were also included, that is, one
station from one nearby subnetwork and another from another nearby subnetwork, exclusive of
the stations in the global subnetwork. All the subnetworks vary day by day depending on the
station data actually available for the day considered. This network strategy has changed
compared to past ULR reanalyses, benefitting from the experience of the Massachusetts
Institute of Technology (MIT) IGS analysis centre.

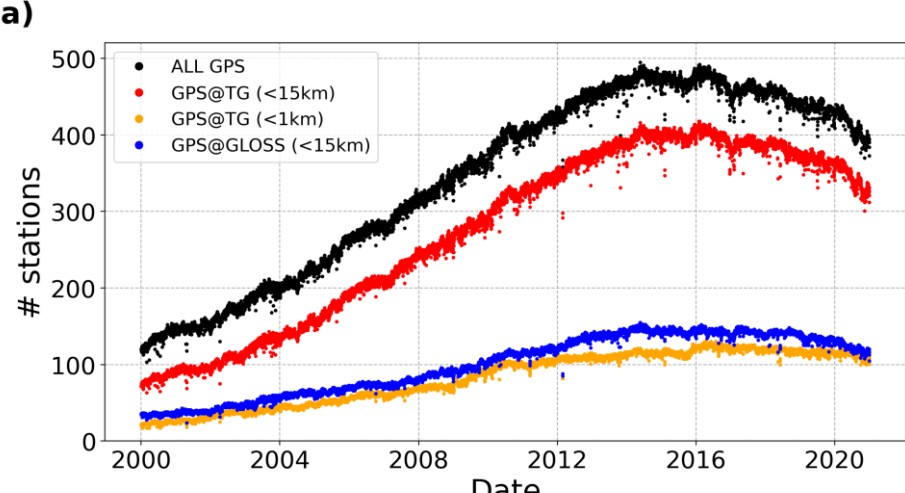


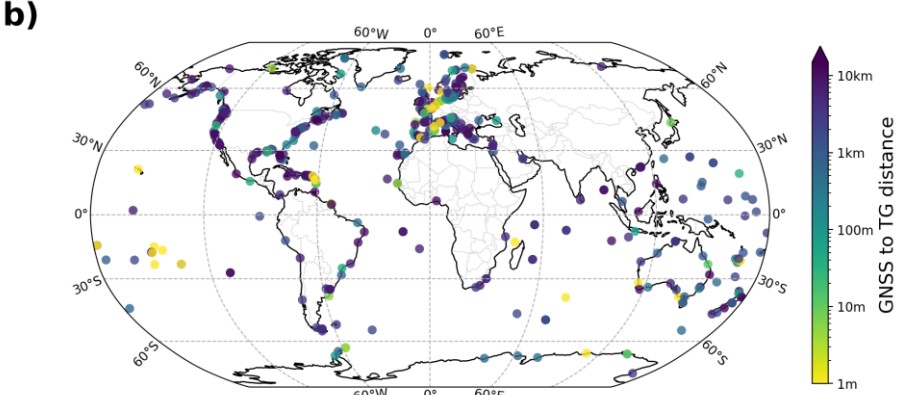

**Figure 2**: a) Evolution of station availability in ULR-repro3 (all in black), within 15 km and 1 km distance from a tide gauge (red and orange, respectively), and GLOSS tide gauge site (blue); b) Spatial distribution of GNSS stations and their distance to tide gauges considered in this study.

The loosely-constrained station positions and tropospheric delays for the common stations and the satellite orbital and Earth rotation parameters estimated from the subnetwork data analyses were combined using GLOBK (Herring *et al*., 2015) to obtain the daily global solutions, which include all stations available each day with their positions expressed in a common but yet undetermined terrestrial frame. These daily global solutions were then stacked into a long-term solution using the CATREF software package (Altamimi *et al*., 2018) with a time-dependent functional model that included translation, rotation and scale transformation parameters between daily and long-term frames, estimated simultaneously with the mean station positions (at the reference epoch 2010.5), annual and semi-annual signals and velocities. The scale parameters, that represent the mean height changes of all the sites, are available upon request, especially for users interested in global sea level rise.

Note that position offset discontinuities (mostly due to equipment changes and earthquakes), as well as station velocity changes and post-seismic displacements, were added to the above modelling, where appropriate. Since experienced analysts still tend to perform better than automatic methods (Gazeaux *et al*., 2013), the position offsets were identified and adopted by expert eyeball using all positioning components (i.e., including north and east components). To facilitate this task, the equipment changes reported in the GNSS station logs were considered, as well as the co-seismic displacements larger than 2 mm predicted with the earthquakes database and modelling by Métivier *et al*. (2014). When a position discontinuity was detected in a time series, the station position was estimated separately before and after the discontinuity together with the offset amplitude. The velocities before and after each position discontinuity

were tightly constrained (0.01 mm/yr), unless a velocity discontinuity was suspected. In the
latter case (less than 2% of the GNSS stations considered in ULR-repro3), no constraint was
applied and different velocities were estimated for each period of data around the discontinuity.

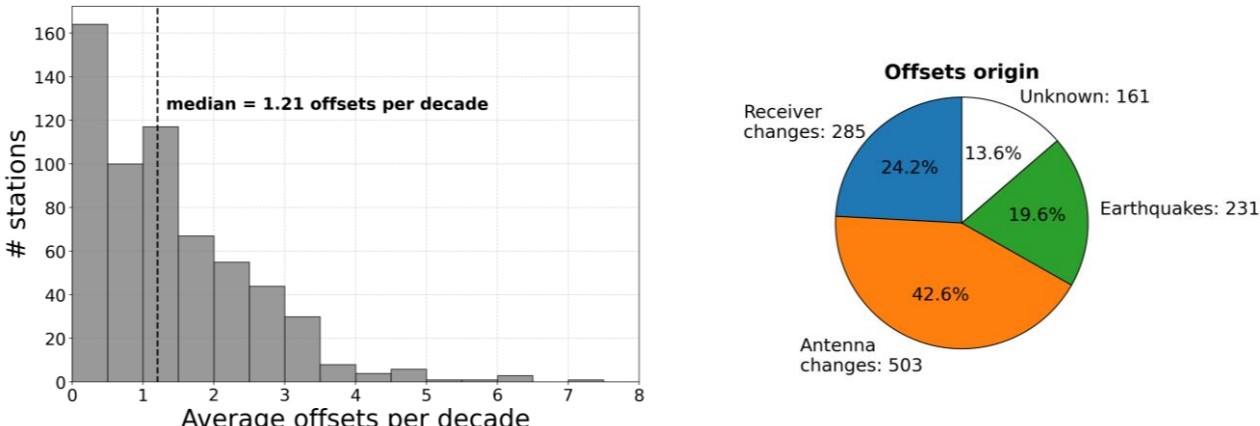

**Figure 3**: Average station position offsets per decade (histogram) and offsets origin (piechart)

The above procedure also included manual editing to identify (and remove) outliers as well as
additional non-documented position offset discontinuities. It was iterated until convergence
(expert eyeball). Overall, 1.2 offset discontinuities were detected per decade and per station,
mostly caused by equipment changes (66.8%) and earthquakes (19.6%), whereas the remaining
13.6% were flagged as unknown (Figure 3) due to the lack of available metadata.

The long-term terrestrial frame, in which the estimated velocities are ultimately expressed, was
finally aligned to the ITRF2014 (Altamimi *et al.*, 2017) by applying minimal constraints to all
the transformation parameters (translation, rotation, scale, and their rates) with respect to the
positions and velocities of a stable subset of about 35 well-distributed reference frame stations.
This step resulted in daily position time series expressed in the ITRF2014 frame for all (601)
stations considered in ULR-repro3. From this set of position time series, only stations having
more than three years between two consecutive position discontinuities and with data gaps not
exceeding 30% were retained for the next step as input (546 stations, among which 161 are
reference stations and 457 are nearby a tide gauge).

*2.2.3. Stochastic modelling and time-correlated noise*
The last step was the estimation of the parameters of interest (primarily here station velocities)
and their uncertainties, where both a functional and a stochastic model were adjusted to each of
the position time series found using the procedure described in Section 2.2.1 on a station by
station basis, following the equation (E1):

$$x(t) = x_{ref} + v_x(t - t_{ref}) + \sum_{i=1}^{N_0} a_i H(t - t_i) + \sum_{j=1}^{3} \left[ s_j \sin\left(\frac{2\pi}{\tau_j} t\right) + c_j \cos\left(\frac{2\pi}{\tau_j} t\right) \right]$$

$$+ \sum_{d=1}^{8} \left[ s_d \sin\left(\frac{2\pi}{\tau_d} t\right) + c_d \cos\left(\frac{2\pi}{\tau_d} t\right) \right] \tag{E1}$$

$$+ \sum_{f=1}^{3} \left[ s_f \sin\left(\frac{2\pi}{\tau_f} t\right) + c_f \cos\left(\frac{2\pi}{\tau_f} t\right) \right] + \sum_{k=1}^{N_{PSD}} PSD_k(t)$$

where $x_{ref}$ is the position at the reference epoch $t_{ref}$, defined arbitrarily as the mid of the
observation period considered (2000.0-2021.0),
$v_x$ is the linear velocity,
$H(t - t_i) = \begin{cases} 0 \ if \ t < t_i \\ 1 \ if \ t \geq t_i \end{cases}$ is the Heaviside function that multiplies the position offset $a_i$
$\tau_j = \frac{1}{j}$ is the period in years of the seasonal term j (annual, semi-annual and ter-annual),
$\tau_d = \frac{P_D}{365.25}$ is the period in years ($P_D$ in days) of the draconitic signals,
$\tau_f = \frac{P_F}{365.25}$ is the period in years ($P_F$ in days) of the fortnightly signals.
$$PSD_k(t) = \begin{cases} a_k \log\left(1 + \frac{t - t_k}{\tau_k}\right) if \ PSD \ model \ is \ log \\ a_k \left(1 - e^{-\frac{t-t_k}{\tau_k}}\right) if \ PSD \ model \ is \ exp \\ a_{1k} \log\left(1 + \frac{t - t_k}{\tau_{1k}}\right) + a_{2k} \left(1 - e^{-\frac{t-t_k}{\tau_{2k}}}\right) if \ PSD \ model \ is \ log + exp \\ a_{1k} \log\left(1 + \frac{t - t_k}{\tau_{1k}}\right) + a_{2k} \log\left(1 + \frac{t - t_k}{\tau_{2k}}\right) if \ PSD \ model \ is \ log + log \\ a_{1k} \left(1 - e^{-\frac{t-t_k}{\tau_{1k}}}\right) + a_{2k} \left(1 - e^{-\frac{t-t_k}{\tau_{2k}}}\right) if \ PSD \ model \ is \ exp + exp \end{cases}$$
In this step, an additional and independent time series editing was considered to eliminate
possibly remaining unreliable estimates from the previous step. The position estimates were
compared to a running monthly median. Any epoch with a position showing a difference from
the median exceeding five times the median absolute deviation in at least one component was
discarded.

The stochastic model considered a linear combination of white noise and power law process
(WN+PL), whose parameters (the stochastic process amplitudes and the spectral index of the
power-law process) were estimated using the Restricted Maximum Likelihood Estimation
method (Patterson & Thompson, 1971; Koch, 1986; Gobron *et al*., 2022). To obtain realistic
stochastic parameter estimates, non-tidal atmospheric loading (NTAL) displacements were also
subtracted from the position time series prior to this adjustment, following the recommendation
of Gobron *et al*. (2021). These NTAL displacements were obtained from the Earth System
Modelling team of the German Research Centre for Geosciences in Potsdam (Dill and Dobslaw,
297 2013).


The functional model included an intercept, a linear trend (velocity), the position offsets
identified in the previous step, three seasonal terms (annual, semiannual, and terannual),
periodic terms at the first eight harmonics of the GPS draconitic year (351.4 days; Ray et al.,
2008), and three fortnightly terms with periods of 13.62, 14.19 and 14.76 days (Penna and
Stewart, 2003; Amiri-Simkooei, 2013). The parameters of this functional model, and their
uncertainties, were estimated using the weighted least squares estimator with the inverse of the
estimated WN+PL model covariance matrix as weight matrix. During the observation time
span, some stations (44) recorded significant co-seismic offsets and transient post-seismic
signals, in which case the modeling was further extended to include velocity changes, and
logarithmic or exponential decay functions according to the observed time evolution.

## 310 2.3 Estimates of vertical land motion

### 311 *2.3.1 Position time series & Vertical velocities*

The GNSS products of primary interest for sea level studies are the station position time series
and vertical velocity estimates as underlined in the founding charter of the IGS working group
"GNSS Tide Gauge Benchmark Monitoring" (Schöne *et al*., 2009) and later on in the
implementation plan of the GLOSS programme (IOC, 2012). In the following, we focus on the
vertical positioning component. However, the horizontal components are made available too,
and can be useful for other geophysical applications. Figure 4 shows the ULR-repro3 vertical
velocity field, and the corresponding uncertainties. This GNSS velocity field ultimately consists
of 546 stations, among which 457 are within 15 km from a tide gauge. This number decreases
to 135 for stations less than 1 km from a tide gauge. Note that the stations inland are IGS
reference frame stations (section 2.1).

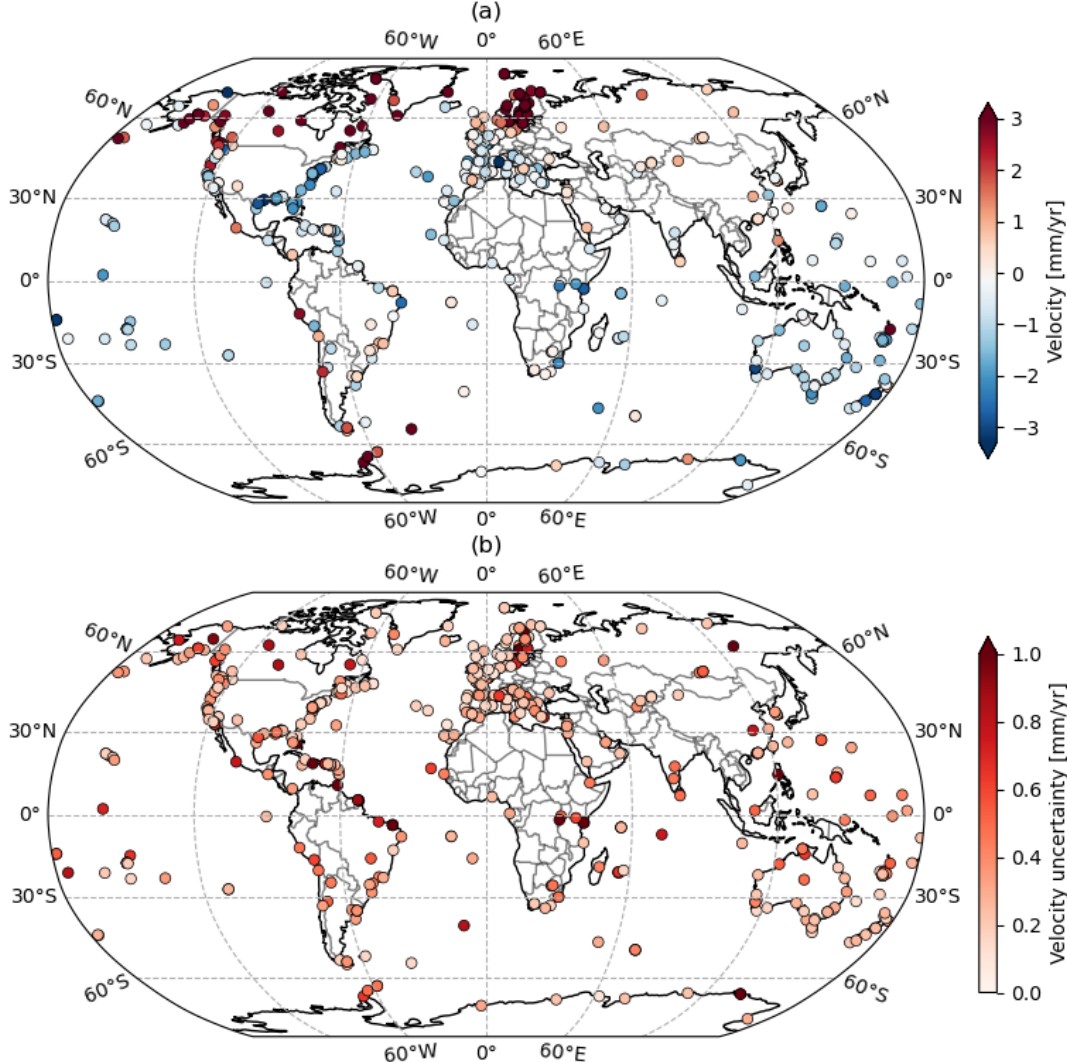

**Figure 4**: Vertical velocities (a), and associated uncertainties (b), estimated for the stations with at least 3 years of continuous measurement (see text).

Overall, the geographical patterns observed in Figure 4.a are consistent with known geophysical processes such as uplift in the northern latitudes of Europe and North America due to Glacial Isostatic Adjustment (GIA), or subsidence along the northern coastlines of the Gulf of Mexico primarily driven by ground water depletion and sediment compaction, also observed in previous and independent GNSS analysis results (e.g., Blewitt *et al*., 2018; Hammond *et al*., 2021). The eight stations with velocity discontinuities are not plotted in Figure 4.

### 2.3.2 Data availability

The ULR-repro3 products are available from the online Digital Object Identifier (DOI) landing page (https://doi.org/10.26166/sonel_ulr7a; Gravelle et al., 2022). That is, the station position

time series together with the estimated velocities for all positioning components (north, east,
and up). These products are hosted at the SONEL scientific service, which serves as data
assembly centre dedicated to GNSS data at tide gauges (Wöppelmann *et al.*, 2021) for the
international GLOSS programme (IOC, 2012). As a UNESCO-related programme, the service
complies with the UNESCO open access data policy (i.e., the data sets are available free of
charge without any barriers) and strives towards providing the highest international standards,
in particular in terms of long-term availability and permanent access. Note that the ULR-repro3
reanalysis yielded other parameter estimates, which can be of interest to other geophysical
applications (e.g., station position offsets related to earthquakes and seasonal signals). These
are also made available via SONEL.

## 347  3 Products quality

## 348  3.1 Average time correlation properties

Previous studies have documented the presence of both power-law noise (PL) and white noise
(WN) in GNSS station position time series (e.g., Williams *et al.*, 2004; Santamaria-Gomez *et*
*al.*, 2011; Gobron *et al.*, 2021; Santamaria-Gomez & Ray, 2021). Such time-correlated
properties are also evidenced by Figure 5 for ULR-repro3, where Lomb-Scargle periodograms
of all detrended station position time series were averaged. As highlighted by the red curve in
Figure 5 (in logarithmic scales), the power-law process induces the negative trend at low
frequencies (that is, a spectral power $\propto 1/f^{\alpha}$), whereas the white noise causes the flattening at
high frequencies. This flattening is especially visible above 22.8 cpy, where the power of the
white noise exceeds that of the power-law process. Note that, on the one hand, the background
shape of the average periodogram (Figure 5) is accounted for by the WN+PL stochastic model,
presented in section 2.2.3, and adjusted to each position time series. On the other hand, the
functional model accounts for the spectral peaks marked by coloured vertical lines, which
correspond to well-identified periodic oscillations common to most GPS solutions (Ray *et al.*,

362     2008).

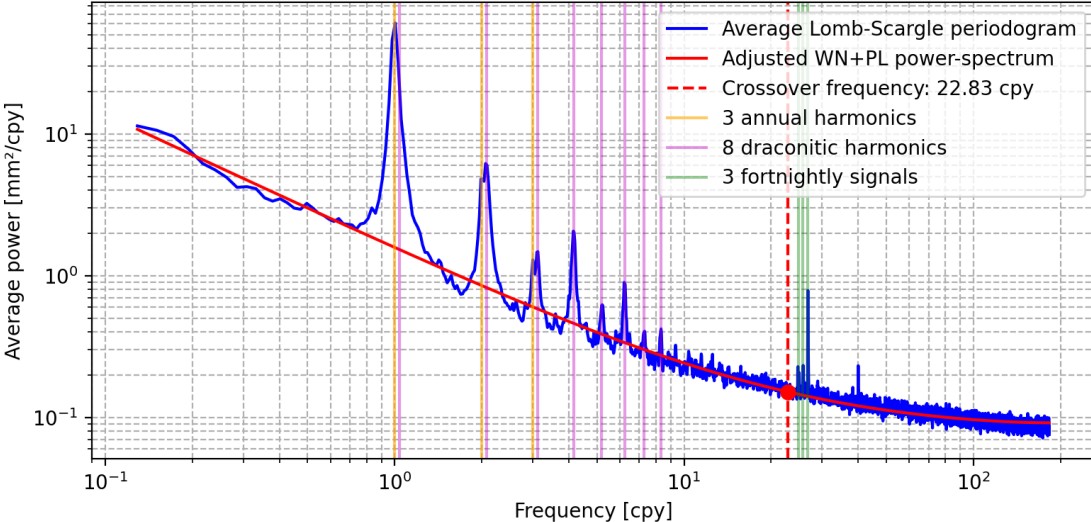

**Figure 5**: Average Lomb-Scargle periodogram for the ULR-repro3 detrended vertical position time series corrected for NTAL displacements (frequency unit is cycles per year or cpy).

## 3.2 Stochastic properties of position time series

The periodogram in Figure 5 does not provide information about the properties of individual stations. By contrast, Figure 6 highlights the stochastic process amplitudes and the spectral index of the WN+PL stochastic models adjusted to the individual vertical position time series.

The median value of the spectral indices is -0.94, that is, close to -1.00, which confirms the prevalence of a flicker-like noise in the low frequency band. The spectral indices show no clear latitudinal dependency (Figure 6.g-h). Since power-law amplitudes depend on the spectral index values, they were transformed into a modified empirical standard deviation (Gobron *et al.*, 2021), expressed in mm, enabling a more rigorous comparison between noise amplitudes and Root Mean Squared Errors (RMSE) values. No latitudinal dependency is revealed in Figure 6.e. By contrast, the white noise amplitudes show largest values within the tropical band (Figure 6.c), and lower values at high latitude, but are mostly non-zero thanks to the NTAL corrections (Gobron *et al.*, 2021). Logically, this pattern also appears in the RMSE (Figure 6.a), as it quantifies the combined influence of the white noise and power-law processes.

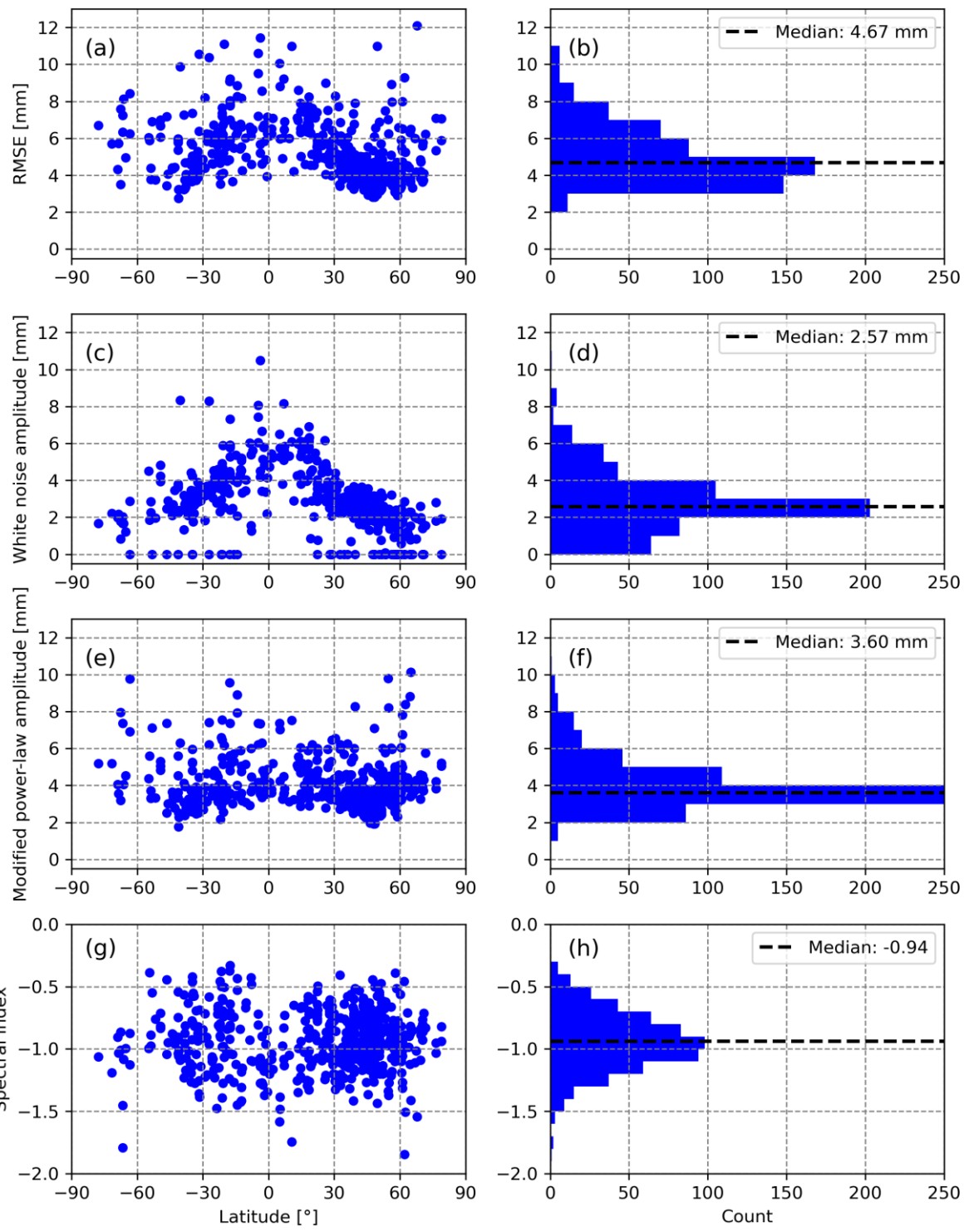

**Figure 6**: Vertical position time series RMSE (a) and (b), white noise amplitudes (c) and (d); modified power-law amplitudes (e) and (f); and spectral indices (g) and (h). See text for details.

## 3.3 Vertical velocity uncertainties

An important consequence of temporally correlated noise in time series of GNSS positions is its impact on the uncertainties of GNSS-derived velocities, which can be largely

underestimated, up to a factor of ten (Williams *et al.*, 2004), if the temporal correlations are
ignored. Figure 7 shows the distribution of the vertical velocity uncertainties obtained for the
ULR-repro3 stations considering the stochastic properties estimated above. Their median value
is 0.27 mm/yr with 83% of the stations displaying a vertical velocity uncertainty below 0.5
mm/yr. The colouring in Figure 7.a indicates that the largest velocity uncertainties typically
correspond to the stations with the shortest records.

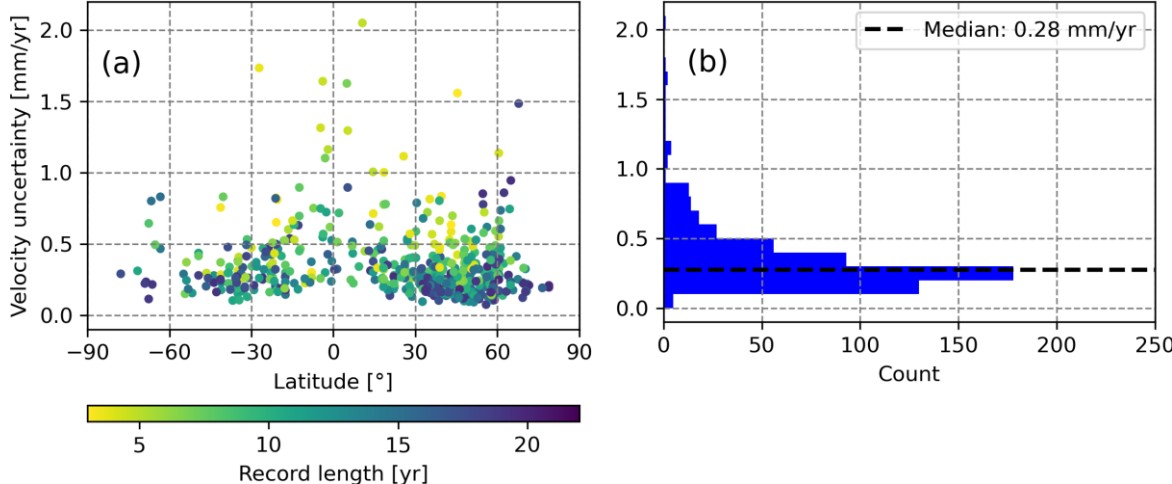


**Figure 7**: Vertical velocity uncertainties as a function of the geographical latitude with colour
corresponding to the record length (a) and histogram (b).

## 3.4 Highlights *wrt* previous ULR reanalysis

To appraise the progress accomplished with the ULR-repro3 reanalysis, the position time series
from the previous reanalysis (Santamaria-Gomez *et al.*, 2017) were retrieved at SONEL, and
the same processing (last step in section 2) was applied for a rigorous comparison (same non-
tidal atmospheric loading corrections, and same functional and stochastic models). This
comparison involved the 251 common stations. Figure 8 indicates a substantial reduction of
28% in the median vertical velocity uncertainties, from 0.35 mm/yr down to 0.25 mm/yr (ULR-
repro3), which is below the uncertainty threshold reported by Griffiths and Ray (2016) using
simulations to investigate the effect of position offsets and record lengthening. However, it is
worth noting that the community interested in monitoring vertical land motion at tide gauges
tend to limit changes in GNSS equipment to the strictly unavoidable (failure, destruction, etc.)
as recommended by the IGS-related working group (Schöne *et al.*, 2009). As a result, the
average return period of offsets is about four years longer here (Figure 3) than observed for the
entire set of stations contributing to the IGS repro3 campaign (Rebischung *et al.*, 2021), hence
partly explaining the improved velocity uncertainties observed with ULR-repro3 reanalysis.

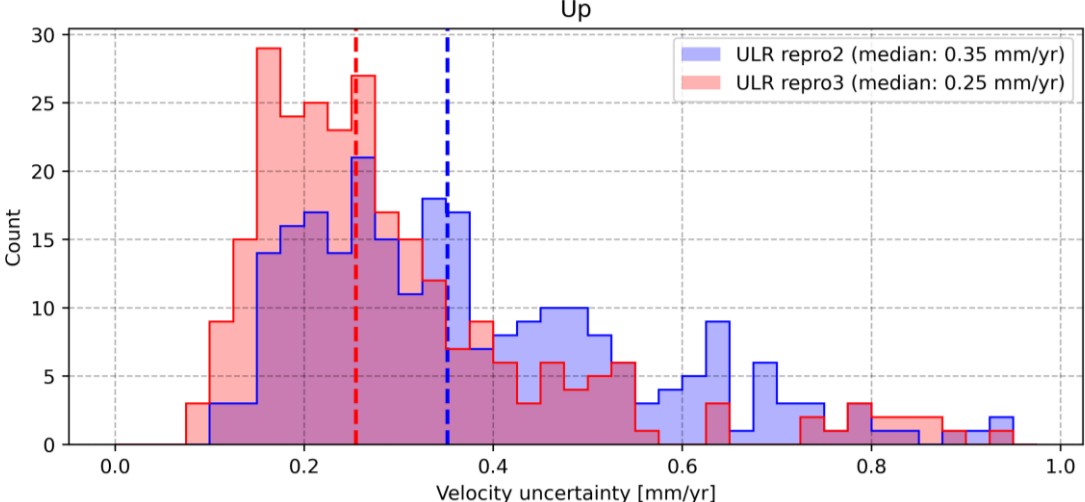


**Figure 8**: Vertical velocity uncertainties for ULR-repro3 wrt the previous ULR solution based on the
251 common stations. The vertical dashed lines correspond to the medians.

The marked improvement in the quality of ULR-repro3 products can also be appraised from
the RMSE position residuals (median value of 5.3 mm down to 4.9 mm now) and the amplitude
of white noise (from 3.4 mm to 2.8 mm), whereas the power-law amplitude and spectral index
remained equivalent (3.8 mm and -0.93, respectively).
In addition to the progress achieved over the previous ULR solution, the quality of the ULR-
repro3 solution was also confirmed by the comparisons undertaken within the IGS reanalysis
campaign, showing that the noise content in ULR-repro3 is comparable to that of most of the
other contributing solutions and analysis centres (Rebischung *et al*., 2021).
## 4 Concluding remarks
This paper has presented the latest GNSS data reanalysis carried out by the ULR group within
the international IGS framework, yielding time series of position estimates to measure the
vertical land motion nearby tide gauges. It includes an increased number of GNSS stations with
an extended time span. Along with the velocity estimates, their uncertainties were obtained by
modelling the temporally correlated noise processes inherent in the data, after correcting the
position time series for non-tidal atmospheric loading displacements, as recommended by
Gobron *et al*. (2021). Overall, the comparisons indicate that ULR-repro3 represents a marked
improvement in its product quality over the previous reanalysis, with a notable reduction in
median vertical velocity uncertainty by 28% (Figure 8).
An interesting perspective will be to examine the differences with global reanalyses obtained
by other groups complying with the latest IGS standards, but using different analyst choices at
any of the major GNSS processing steps described in section 2 (e.g., Blewitt *et al.*, 2016;
Männel *et al.*, 2022). A related perspective will be to address the issue of which reanalysis is
best for the non-expert sea level user, if any (Ballu *et al.*, 2019). In this respect, the Commission
on Mean Sea Level and Tides from the International Association for the Physical Sciences of
the Oceans (IAPSO) could provide a stimulating framework to gather experts and users
worldwide, and reflect on the issue posed by multiple high-quality GNSS reanalyses, as IAPSO
did nearly thirty years ago when the issue of geodetic fixing of tide gauge benchmarks was
considered with the advances of space geodesy (Carter *et al.*, 1994).

**Acknowledgments**
This study was financially supported by the CNRS/INSU research agency via the SONEL &
RENAG observation systems. Our study benefited tremendously from agencies making their
data available to IGS and SONEL. The authors would like to acknowledge the crucial role
played by the high-performance computing centre of La Rochelle University, especially with
Mozart & Thor supercomputers. Part of this research were made possible by support from
NASA grant 80NSSC18K0457 and NSF grant NSF-IF-1843686. We are grateful to Laurent
Métivier for providing the modelled earthquake displacements and Jean-Paul Boy for the ocean
tidal loading corrections. Finally, we would like to thank both anonymous reviewers for their
comments that contributed to improve the paper.

**Author contributions**
The project was defined by MGr and GW as a contribution to the third reprocessing ('repro3')
campaign of the International GNSS Service (IGS). MGr processed the GPS data with support
from TH (GAMIT/GLOBK software packages, network design, and strategy for subnetwork
design and orbit adjustment), ZA (CATREF software and reference frame alignment), PR
(product quality assessment within 'repro3'), and MGu (strategic use of the high-performance
computing centre). KG prepared the NTAL corrections, assessed the stochastic properties of
the time series and produced the final velocity field. All authors contributed to the analysis and
discussion of the results. The first manuscript draft was written by GW. All authors contributed
to the subsequent versions, and approved the final manuscript.

**Competing interests**
The contact author has declared that none of the authors has any competing interests.

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
