# Peer review of "The ULR-repro3 GPS data reanalysis and its estimates of vertical land motion at tide gauges for sea level science"

_Earth System Science Data, 2022_

## Referee Comment (RC2)

Review of Gravelle et al.:
The ULR-repro3 GPS data reanalysis and its estimates of vertical land motion at tide gauges for sea level science (essd-2022-235)

Gravelle et al. present a new reanalysis of GNSS data, focusing on stations that are collocated with tide gauges.  The authors first describe how the input dataset was chosen and then provide a detailed account of how the GNSS data was processed. The steps taken were in accordance with the international standards adopted by the IGS for the third reprocessing campaign. An analysis of the sources of uncertainty in the data products especially relevant to the sea level community (vertical positioning and vertical rates), and the geographic variability therein, is also presented. The authors conclude with a convincing demonstration that the reanalysis provides an improvement over the previous reanalysis campaign. It is clear that a great deal of effort went into the creation of this dataset, and it is a welcome addition. The authors should especially be applauded for their work in making the data accessible; the data products hosted at the SONEL scientific service are available free of charge and without barriers.

This is a timely paper, and in my opinion, it should proceed with minor revisions.

**Major comments:**
The purpose of the paper is to present the GNSS data reanalysis of vertical land motion nearby tide gauges, and, quite correctly, the discussion primarily focuses on the GNSS analysis. Little information, however, is given about the tide gauge data. Could the authors indicate where the tide gauge data could be accessed? Will information available on the SONEL archive only relate to the GNSS and that on the GLOSS archive only relate to the tide gauges? When giving the GNSS station information, will an identifier for the nearby tide gauge be included? A map showing the spatial distribution of GNSS station distance to tide gauges could be useful, perhaps in supplementary material (in addition to the information presented in Figure 2).

Section 2.2.3 Stochastic modelling and time-correlated noise
The equation for the station position is given on the About page of https://www.sonel.org/-Vertical-land-movements-.html. I suggest having this information in the paper as well.

**Minor Comments**

**Title**
Is there a reason why GPS was chosen for the title? GNSS is used almost exclusively elsewhere in the main text.

**Abstract**
Please define GNSS.

**Main Text**

L56: use of semicolon is grammatically incorrect here; the clause starting with "that is" would not qualify as a stand-alone sentence.

L66: "that" → which

L66: Although the citation for the modelling and corrections adopted for repro3 is given, a short synopsis may also be useful here. Section 2.2.1 and Table 1 do cover this information, so perhaps a shortened version could be given.

One possible edit:

This paper describes the latest ULR solution in a series, complying with the modelling and corrections adopted for 'repro3' (Rebischung, 2021; http://acc.igs.org/repro3/repro3.html), which succeeds previous releases (Wöppelmann *et al*., 2009; Santamaria-Gomez *et al*., 2017).

→

This paper describes the latest ULR solution in a series, succeeding previous releases described in Wöppelmann *et al*., (2009) and Santamaria-Gomez *et al*. (2017). This solution complies with the modelling and corrections adopted for 'repro3' (Rebischung, 2021; http://acc.igs.org/repro3/repro3.html), for example, corrections are made for antenna phase center and solid Earth tides (see Section 2.2.1).

L88: Could you clarify how near to a tide gauge a station must be to satisfy the selection criterion? I suspect it is <=15 km, but this is not explicitly stated.

L94 GLOSS is defined, but it would be useful to have additional information on this program, for example, what data products are made available by it.

L105: Please indicate how many of the 601 stations are reference frame stations.

L112: suggest not repeating "from GNSS measurements" twice in the sentence.

L145-148: This sentence is difficult to parse at first read through; perhaps it could be split into two. What do the authors mean by "converted from relative to absolute"?

L174. Suggest moving parenthetical information to a separate sentence.

L185: hydrologic?

L225: experimented analyst? Do the authors mean experienced?

L240: suggest a comma after "Overall"

L249-251: How many stations satisfy these conditions? How many were reference frame stations vs stations near tide gauges?

L254-258: step is used four times in two sentences, and it is not clear at first read-through whether the authors are referring to a step in the overall procedure or referring to a previous iteration. One possible means of clarification: "a functional and a stochastic model were adjusted to each

of the position time series from the previous step on a station by station basis." → "a functional and a stochastic model were adjusted to each of the position time series found using the procedure described in Section 2.2.1 on a station by station basis."

L286: should this be "of the vertical component"?

L300: suggest "America" changed to either "North America" or "Canada"

L303: Is there a reason the authors used GPS here rather than GNSS?

L304: How many stations are not plotted?

L321-329: Point of clarification, does the power-law and white noise discussed in this section correspond to the noise discussed in section 2.2.3? In general, more description on how to interpret Figure 5 and what details are included on the figure would be welcome.

L335: Perhaps this sentence could be split into two for clarity.

L347: should this be "but *are* mostly non-zero"?

L378: should strict be strictly?

L388: Was there also improvement seen in the North & East components? If so, by how much?

L401: product's?

L402: suggest stating that it is the vertical velocity that experienced the reduction in uncertainty.

**Tables**
Table 1
In the second column suggest writing out Earth Orientation Parameters tide model from Desai and Sibois (2016) as opposed to just the reference.

**Figures**

Fig. 1
Why are some station circles different sizes? If size as well as color corresponds to station duration, a key would be useful.
Why does the record length range from 3 months to 21 years, wasn't there a selection cutoff of >3years? Are these shorter duration stations all French GNSS stations and/or reference frame stations?

It may be useful to have subpanels with regions of higher concentration of stations, e.g. Europe, Western North America, Eastern North America. Or showing the regional subnetworks mentioned in the main text.

Fig. 2
Please label the x axis.
Suggest having the label for all be "ALL GPS" or "ALL GNSS" to make it clear that the GLOSS tide gauges are not included in the tally.

Fig. 3
Could you increase the text size for the piechart labels?
Perhaps change the color corresponding to the unknown category from red to purple to increase the color contrast for colorblind readers
Would also be useful to have a title for the piechart (e.g., "Offsets origin") to avoid needing to reference the caption.

Fig. 4
Panel a -
Are there values in excess of 3 mm/yr? If so, please add triangles to the colorbar to indicate saturation at +/- 3mm/yr.
The stations with vertical velocities near 0 mm/yr are difficult to see. Stations could be outlined in black, or the colormap could be blue yellow red instead.

Panel b -
suggest switching colormaps to a sequential rather than diverging map. In particular, having the same red to blue colormap as in the above panel risks the reader thinking the panels are on the same scale.

Fig. 7
The colorbar for panel a might be better placed below the figure to avoid mistaking "record length [yr]" for the title of the panel.

Fig. 8
It might be easier to read the histogram if the bar graph is filled in with transparent colors.

---

## Author Response (AR1)

**RC1**

Review of the Paper:

The topic important and the paper is very useful for the sea level community. It gives a condensed overview about the new ULR GNSS(GPS) solution for sites at or near tide gauges as part of the IGS reprocessing effort. The paper is well written and easily to understand also for non GNSS-experts. The data is available at a data repository.

*The positive review is appreciated. Thanks.*

A shortfall of paper is the analysis of the results, which is only done against ULR's previous version. The latter has a different time span and short overlap. The comparison with own previous reprocessing is helpful but may not reveal problems associated with the general processing strategy or setup. Since ULR participated in the IGS reprocessing campaign, I assume there are more in-depth analyses of the ULR contribution against other contributors. Rebischung et al. 2021 (EGU and AGU) are slides of presentations and are less helpful to gain confidence in the new ULR solution. Years ago, Deng et al. (https://doi.org/10.1007/1345_2015_156, unfortunately not open access) did a comparison of their processing with the (older) ULR analysis providing, which may also be a way to evaluate this new data set. I strongly encourage the authors to perform similar studies using external solutions and provide results within the paper.

*There are several reasons that raise some concern in following the suggestion:*

*1. As the reviewer mentioned, such a comparison has already been done and presented by Rebischung et al. (2021), and more recently again at AGU in 2021 and at the IGS workshop in 2022. The comparisons showed that the quality of the ULR-repro3 solution is in line with that of most of the other analysis centres (see Figure R1 below).*

*Figure R1: Smoothed daily median formal errors of station positions (Rebischung et al. 2021). ULR solution is within the boundary of the best and "worst" solutions.*

[Figure]

2. *ULR-repro3 is the only solution formally (within IGS) dedicated to the analysis of coastal GNSS stations nearby tide gauges (GFZ was in repro2, but not in repro3, unfortunately). Consequently, its publication would be useful even though other solutions were proven to be less noisy.*

*Figure R1 above shows, however, that there are no obvious problems associated with the processing strategy or setup. In order to provide additional evidences that the ULR-repro3 solution does not suffer from any major processing issues, we compared it with the IGS-repro3 combined solution. In particular, we compared using the RMSE and the Lomb-Scargle periodogram (Figures R2 and R3). As expected, both the RMSE and Lomb-Scargle show that the repro3 combined solution is less noisy than the ULR-repro3 solution. However, the level of noise ULR-repro3 solution remains low (as most, Figure R1) and does not highlight any major estimation issue.*

*Figure R2: RMSE Comparison of the ULR-repro3 and IGS-repro3 combined solutions*

[Figure]

*Figure R3: Lomb-Scargle periodograms of the ULR-repro3 and IGS-repro3 combined solutions*

[Figure]

*Considering our comments above and that a dedicated article is in preparation by the leaders of the IGS-repro3 combination and comparison campaign, we have followed the suggestion with a short summary paragraph at the end of Section 3.4.*

In general, I recommend to publish the paper with minor corrections.

Comments which may help the authors to further improve the paper:

Title: the term GPS is used, but later consequently GNSS, better have consistent naming

> *The term GPS in the title is intended to make clear what type of GNSS data was analysed. The text then uses GNSS consistently, except to specify the GNSS-type (Section 2.1) or where a GPS-technique feature is specifically addressed.*

Line 46: I assume the GNSS orbit estimate is CoM, but is this also true for GNSS derived coordinates and velocities?

> *Once the alignment to the ITRF2014 is done, GNSS positions and velocities are expressed relative to the origin of the ITRF2014, which follows the CoM (as observed by Satellite Laser Ranging or SLR) over long-term time scales, but at short time scales it corresponds to the Centre of Figure (CoF). Thus, velocities (long-term by definition) are expressed relative to the CoM. However, daily positions are relative to CoF. See details in Dong et al. (2003) in JGR (Origin of the International Terrestrial Reference Frame) and Altamimi et al. (2014) in JGR for the case study of ITRF2014.*

Line 89: Although the authors using "continuous" only here and Fig.4, most of the Geoscientist may understand this term as the opposite to "campaign" GNSS. Also, the two citations using this term refer to it as long-term. I suggest another wording.

> *True that some geodesists may understand "continuous" as permanent station observations in contrast to "campaign" as episodic observations. We did not find a better wording. This is why we clarify the term right after using it, and we do not use it anymore, but in Figure 4 which asks the reader to refer to the text. If you have a better suggestion, we will be happy to consider it.*

Line 98: is 3 months correct? For me this sounds contrary to the statements in line 86ff.

> *The short record length (3 months) stations correspond to French stations, whose GNSS was recently installed to fulfil the requirements of the GLOSS program. This is clarified in the text just before introducing the GLOSS programme, and providing additional information requested by Reviewer #2.*

Line 99: Santamaria-Gomez et al., 2017: the supplementary material says 757 stations with data between 1995.0 and 2015.0

- here the authors wrote 2013.

  *There was a typo in Santamaria-Gomez et al. We confirm that the last year processed was 2013 (in other words, from 1995.0 to 2014.0).*

- any statement, why the authors processed less stations

- any statement, why this solution starts later than their previous reprocessing effort?

  *The main reason was to cope with the deadlines of the IGS-repro3. We did not have the resources to extend back in time before 2000 (CPU cluster access was challenging), nor was this a requirement for the IGS reanalysis campaign. For instance, the WHU group solution extends back to 2008, and the MIT or GRG groups back to 2000, as we ultimately did. The manual editing of the time series was also a long time-consuming step as we did it carefully (carried out on the three components).*

  *We therefore were stricter in the station selection criteria, rejecting more stations than in the previous reprocessing effort. Hopefully, this will change for the next reanalysis campaign (new cluster, increased automation in quality check procedures...).*

Line 113: could you specify the terminus "many corrections", likely for the supplementary material

  *For clarity, we have changed this with the explicit section number, where the details can be found. We have prepared a Table with rather technical details for GPS specialists. However, we consider this of little use for the vast majority of the readers/users of the data paper. We attach the Table to our response, and if the reviewer or editor insist, we will add it as supplemental material to the paper.*

Line 138/Table1: which IONEX files are used, IGS combined?

  *Yes, indeed, the IGS combined IONEX files are used. It is now stated in Table 1.*

Line 138/Table1: FES2014b also contains Ssa and Sa.

  - Are you truncating this model, and if yes, to which tides; what leakage do you expect?
  - How (if) does Ssa & Sa map to the spectral behavior of your orbits (and coordinates). Since a correction for tides are sensed by both, satellites and tide gauges, this would help the user to understand possible side effects.

  *We did not truncate FES2014b model, neither for the station displacements nor for the variations in the gravity field. The effects of the ocean tides are thus corrected the best it can using the state-of-the-art, including Sa and Ssa.*

  *FES2014b modelling errors can indeed exist and propagate into ULR solutions (and likely contribute peaks at ~14 days). The question of whether possible errors in Sa and Ssa in*

*FES2014b model yield significant errors in ULR7 solution, and which are these errors, is out of the scope of our data paper.*

Line 195: pls specify "several" and supply plots with the subnetworks in supplementary material.

*Several is up to 10. Now added in parenthesis. Figure S1 (added as Supplemental Material) illustrates the subnetworks distribution for a given (random) day 2018-01-01.*

Line 195: The subnetworks are fixed though 2000.0 till 2021.0? or vary day by day?

*The subnetworks vary day by day. It is now specified in the text.*

Lines 198 to 205: This section needs some more explanations. For those not familiar with GNSS processing it is interesting to understand how the alignment and transformation of the sub-networks is performed, but I failed to understand the concept.

*These lines and paragraph refer to a practical aspect of computational efficiency. The Figure S1 now added in the Supplemental Material (suggested above) will likely help illustrating this aspect of station distribution into sub-networks.*

*The combination of the sub-networks and alignment are explained in the next paragraph, and the very last of the section. They refer to the literature for the daily combination (Herring et al., 2015) and then for the frame alignment / transformation to Altamimi et al. (2018). We provide information on the choices that we adopted for our specific case study that may help the reader with geodetic background. Further details on the geodetic concepts can be found in those manuals. Therein, the reader can find additional references. For instance, the GLOBK algorithms for network combinations are explained in Dong D., T. A. Herring, and R. W. King, Estimating Regional Deformation from a Combination of Space and Terrestrial Geodetic Data, J. Geodesy, 72, 200–214, 1998. There are two concepts: You can estimate the rotation and translation with least-squares fitting of the coordinates of the common stations or you can construct the site coordinate covariance matrix that allows for rotation and translation of the network and simply combining the networks allows the systems to re-orient without any explicit rotation and translation parameters. The latter is a more advanced concept in estimation theory but one which is used in GLOBK extensively.*

Line 230ff: pls give some more information about the handling at earthquake sites. - How many days are used to estimate offsets; - any outlier control of the daily solutions, - how you handle postseismic deformation; - what, if new earthquakes occur within your fitting period?

*The offsets are estimated during the stacking process with CATREF software. The offset dates are given by the user in a specific file. The software then estimates the amplitude as the difference between the average of the detrended positions before and after the offset. Post-seismic signals were first detected visually, then corrected using the IGS estimates, before a new stacking iteration is performed.*

Line 292: The data doi web site say 554 stations, while here only 546 are mentioned

*It is now corrected in the doi web page, thanks for pointing this out.*

Line 330/Fig.5: what causes the peak near 7*10^1cpy

*We do not see any peak at this frequency (70 cpy). Note that we see a peak at 40 cpy, which was already present in the previous (repro2) ULR solution as well in MIT solution. The peak is still present in ULR and MIT repro3 solutions. Its origin remains unknown, it does not correspond to a tidal aliasing or to another known process. Further research will be needed to identify its origin.*

Line 368ff: Did you perform the comparison for 2000.0 – 2013(15).0?

*No, we compared the results obtained by considering the entire span of each dataset so that the resulting differences highlight the progress that a user can expect by using the new ULR-repro3 solution, whatever its origin (record length or advanced corrections and modelling).*

Some (likely) typos

Line 23: university -> University

*Corrected*

Line 83: RINEX: any link to or citation of?

*A link is now added (https://igs.org/wg/rinex/)*

Line 213: Herring et al., 2021 is missing in the reference list

*The year is corrected as in Ref 24 adding the online link to whole documentation.*

Line 225: experimented or experienced?

*Experiences (corrected now).*

Line 240: per decade and station?

*Yes, per decade and per station, it is specified now*

Line 265: is Gobron et al. in press: Gobron, K., Rebischung, P., de Viron, O. et al. Impact of offsets on assessing the low-frequency stochastic properties of geodetic time series. J Geod **96**, 46 (2022). https://doi.org/10.1007/s00190-022-01634-9? Or a different paper?

*Updated.*

Line 269: correct research center for geosciences -> Research Center for Geo…

*Done.*

Line 408: CMSLT should all be upper case letters

*Done.*

| Products | SINEX | ULR0R03FIN_yyyyddd000000_01D_01D_SOL.SNX.gz |
|---|---|---|

| Observables | | |
|---|---|---|
| GNSS (included from) | GPS (2003/01/01) |
| Observable types | Doubly differenced phase (GPS: L1&L2) and code observations |
| Sampling rate | 2 minute, 30-second cleaning |
| Elevation cutoff | 10° |
| Elevation-dependent inverse weights (sigma² = ) | sqrt(a^2+b^2/sin(e)^2)); a and b are site and day dependent. (5.5 and 3.5 mm typical) |
| Data span | 24 hr |

| Measurement corrections | | |
|---|---|---|
| Code biases | C1-P1 biases from CODE monthly values |
| Phase biases | DD solution |
| RHC phase rotation corrections | Wu et al., 1993 |
| Satellite antenna-centre of mass offsets | igsR3_2135.atx |
| Satellite antenna phase variations | igsR3_2135.atx |
| Ground antenna phase center corrections | igsR3_2135.atx |
| Radome phase center corrections | Applied if in igsR3_2135.atx (NONE if not) |
| Marker => antenna ARP eccentricity | dN, dE, dU eccentricities from site logs (mx header if no site log available) |
| Relativistic corr. for satellite clocks (-2*R*V/c) | Applied |
| Gravitational bending | IERS2010 |

| Terrestrial Reference Frame | | |
|---|---|---|
| Terrestrial reference frame | |
| Alignment of SINEX solutions | NNR |
| Orbits consistent with SINEX? | Yes |
| Alignment of clock solutions | No |

| Inertial Frame | | |
|---|---|---|
| Precession / nutation | IAU2000A |
| A priori EOPs | Bulletin A |
| Tidal variations | Desai & Sibois (2016) |
| UT1 Libration | IERS 2010 |

| Geopotential | | |
|---|---|---|
| Static gravity field | EGM2008 12x12 |
| Non-tidal variations | |
| Solid Earth tides | IERS2010 |
| Ocean tides | FES2014b |
| Solid Earth pole tide | IERS2010 |
| Ocean pole tide | IERS2010 |
| Mean pole model | Linear mean pole |

| Station displacements | | |
|---|---|---|
| Solid Earth tides | IERS2010 |
| Ocean tidal loading | FES2014b |
| Ocean tidal loading center of mass corr. | Applied to orbits |
| S1/S2 atmospheric loading | None |
| S1/S2 atmospheric loading CM corr. | None |
| Solid Earth pole tide | IERS2010 |
| Ocean Pole Tide | IERS2010 |
| Mean pole model | Linear mean pole |
| Non-tidal loading atmospheric pressure | None |
| Non-tidal loading ocean bottom pressure | None |
| Non-tidal loading surface hydrology | None |
| Non-tidal loading (general remarks) | None |

| Orbit dynamics | | |
|---|---|---|
| Arc length | 24 h |
| Shadow zones | Earth and Moon |
| GPS attitude in daylight | Nominal, Kouba model |
| GAL attitude in daylight | Nominal |
| Third bodies | JPL DE405, Sun, Moon, Jupiter, Venus |
| Earth reflected (visible) radiation | Rodriguez-Solano 2009 |
| Earth emitted (infrared) radiation | Rodriguez-Solano 2009 |
| Antenna thrust | IGS METADATA SINEX |
| Thermal re-radiation | |
| Relativistic effects dynamical correction | IERS2010 |
| Relativistic effects gravitational time delay | IERS2010 |
| A priori solar radiation pressure | Direct only |
| Empirical accelerations (& constraints) | ECOM2 with stochastic constraints and selected terms based on obit overlaps and all estimates |
| Stochastic pulses (& constraints) | None |

| Tropospheric delays | | |
|---|---|---|
| Met data | VMF1 |
| A priori zenith delays | VMF1 |
| Adjusted zenith delays (& constraints) | Peicewise linear (1-hr knots) |
| Mapping functions for a priori delays | VMF1 |
| Mapping function for adjusted delays | VMF1 wet |
| Mapping function coefficients | VMF1 |
| A priori gradients | None |
| Adjusted gradients (& constraints) | NW/EW linear over day (apriori constraints) |
| Mapping function for gradients | Chen and Herring (1997) |

| Ionospheric delays | | |
|---|---|---|
| 1st-order effect | Dual frequency LC |
| 2nd-order | IGRF13, TEC from IGS IONEX |
| 3rd-order | Applied |

| Estimated parameters & constraints | | |
|---|---|---|
| Software | GAMIT/GLOBK 10.71 |
| Adjustment method | weighted least-squares+Kalman filter |
| Station coordinates | All station coordinates adjusted; NNR constraints |
| Earth orientation parameters (EOP) | XY offset and rates, LOD with UT1 from integrated LOD (Bulletin A, start of week) |
| Geocenter | No |
| Troposphere zenith wet delays | Piecewise linear (1-hr knots) |
| Troposphere gradients | Piecewise linear (1-hr knots) |
| Ionosphere | None |
| Orbits | 6 Keplerian elements (independent each day). 3 constant radiation parameters, 2/6 twice/once/four per rev direct, once-per-rev B axis (satellite/week dependent). |
| Satellite attitude | nominal yaw |
| Satellite antenna offsets | Yes; tightly constrained |
| Ambiguities | Double difference, Wide and Narrow lanes, Melbourne-Wubbena for WL |
| Satellite clocks | Re-constructed approximately from double difference |
| Receiver clocks | Re-constructed approximately from double difference |
| Satellite code biases | not estimated |
| Satellite phase biases | not estimated |
| Receiver code biases | not estimated |
| Receiver phase biases | not estimated |

**RC2**

Gravelle et al. present a new reanalysis of GNSS data, focusing on stations that are collocated with tide gauges. The authors first describe how the input dataset was chosen and then provide a detailed account of how the GNSS data was processed. The steps taken were in accordance with the international standards adopted by the IGS for the third reprocessing campaign. An analysis of the sources of uncertainty in the data products especially relevant to the sea level community (vertical positioning and vertical rates), and the geographic variability therein, is also presented. The authors conclude with a convincing demonstration that the reanalysis provides an improvement over the previous reanalysis campaign. It is clear that a great deal of effort went into the creation of this dataset, and it is a welcome addition. The authors should especially be applauded for their work in making the data accessible; the data products hosted at the SONEL scientific service are available free of charge and without barriers.

This is a timely paper, and in my opinion, it should proceed with minor revisions.

> *We appreciate this supportive summary.*

Major comments:

The purpose of the paper is to present the GNSS data reanalysis of vertical land motion nearby tide gauges, and, quite correctly, the discussion primarily focuses on the GNSS analysis. Little information, however, is given about the tide gauge data. Could the authors indicate where the tide gauge data could be accessed? Will information available on the SONEL archive only relate to the GNSS and that on the GLOSS archive only relate to the tide gauges? When giving the GNSS station information, will an identifier for the nearby tide gauge be included?

> *As requested later on (Minor comments below) we now provide additional information on the GLOSS programme, which answers these questions. Briefly, GLOSS programme comprises five global data centres, SONEL being the one dedicated to the GNSS data, whereas the other four address different products from tide gauges. Efforts are undertaken to favour interoperability between these data centres, in particular we have succeeded to link the GNSS station information with that of its co-located tide gauge. For example, an identifier and associated URL link in the GNSS station information at SONEL is pointing to the tide gauge station at the PSMSL (GLOSS global data centre for mean sea levels), and vice-versa.*

A map showing the spatial distribution of GNSS station distance to tide gauges could be useful, perhaps in supplementary material (in addition to the information presented in Figure 2).

> *We have added a panel (map) to Figure 2 as suggested with the spatial distribution of GNSS stations and their distance to tide gauges.*

Section 2.2.3 Stochastic modelling and time-correlated noise

The equation for the station position is given on the About page of https://www.sonel.org/-Vertical-land-movements-.html. I suggest having this information in the paper as well.

> *The equation of the model has been added to the text in section 2.2.3*

Minor Comments

Title

Is there a reason why GPS was chosen for the title? GNSS is used almost exclusively elsewhere in the main text.

*The reason is explained in the first line of Section 2.1 "Input data", that is, the title underlines that only GPS data were used, other GNSS (Galileo, Beidou…) were not considered. We can add "GNSS (GPS)" in the title, if the reviewer or the editor finds worth adding a second abbreviation, but this (GNSS) is implicit as soon as GPS is mentioned.*

Abstract
Please define GNSS.

*Done. GNSS is now defined in the abstract.*

Main Text

L56: use of semicolon is grammatically incorrect here; the clause starting with "that is" would not qualify as a stand-alone sentence.

*Semicolon now replaced by a comma.*

L66: "that" -> which

*Done.*

L66: Although the citation for the modelling and corrections adopted for repro3 is given, a short synopsis may also be useful here. Section 2.2.1 and Table 1 do cover this information, so perhaps a shortened version could be given.

One possible edit:

This paper describes the latest ULR solution in a series, complying with the modelling and corrections adopted for 'repro3' (Rebischung, 2021; http://acc.igs.org/repro3/repro3.html), which succeeds previous releases (Wöppelmann et al., 2009; Santamaria-Gomez et al., 2017).

->

This paper describes the latest ULR solution in a series, succeeding previous releases described in Wöppelmann et al., (2009) and Santamaria-Gomez et al. (2017). This solution complies with the modelling and corrections adopted for 'repro3' (Rebischung, 2021; http://acc.igs.org/repro3/repro3.html), for example, corrections are made for antenna phase center and solid Earth tides (see Section 2.2.1).

*Done. Thanks for the suggestion.*

L88: Could you clarify how near to a tide gauge a station must be to satisfy the selection criterion? I suspect it is <=15 km, but this is not explicitly stated.

*Correct (15 km). Now stated in the text.*

L94 GLOSS is defined, but it would be useful to have additional information on this program, for example, what data products are made available by it.

*We have added additional information on GLOSS, which should also answer the questions raised above in Major comments.*

L105: Please indicate how many of the 601 stations are reference frame stations.

*The number of reference stations (176) is now added.*

L112: suggest not repeating "from GNSS measurements" twice in the sentence.

*The repetition (second) is now removed.*

L145-148: This sentence is difficult to parse at first read through; perhaps it could be split into two. What do the authors mean by "converted from relative to absolute"?

*We agree, and have split the sentence into two. The relative aspect refers to an antenna calibration relative to an antenna with an absolute calibration. This is now clarified in the text.*

L174. Suggest moving parenthetical information to a separate sentence.

*It reads better indeed. Done.*

L185: hydrologic?

*Corrected.*

L225: experimented analyst? Do the authors mean experienced?

*Yes! It is corrected now.*

L240: suggest a comma after "Overall"

*Done.*

L249-251: How many stations satisfy these conditions? How many were reference frame stations vs stations near tide gauges?

*546 stations satisfy these conditions, among which 161 are reference stations and 457 are nearby a tide gauge.*

L254-258: step is used four times in two sentences, and it is not clear at first read-through whether the authors are referring to a step in the overall procedure or referring to a previous iteration. One possible means of clarification: "a functional and a stochastic model were adjusted to each of the position time series from the previous step on a station by station basis." -> "a functional and a stochastic model were adjusted to each of the position time series found using the procedure described in Section 2.2.1 on a station by station basis."

*Thanks for the suggestion. Done.*

L286: should this be "of the vertical component"?

*"velocity estimates on the vertical component" has been replaced by "vertical velocity estimates"*

L300: suggest "America" changed to either "North America" or "Canada"

*"America" changed to "North America".*

L303: Is there a reason the authors used GPS here rather than GNSS?

*GPS changed to GNSS.*

L304: How many stations are not plotted?

*There are 8 stations with velocity discontinuities. It is now indicated in the text.*

L321-329: Point of clarification, does the power-law and white noise discussed in this section correspond to the noise discussed in section 2.2.3? In general, more description on how to interpret Figure 5 and what details are included on the figure would be welcome.

*Yes, the white noise and power-law noise presented here are those mentioned in Section 2.2.3. In practice, Figure 5 highlights deterministic and stochastic features that are accounted for by the functional and stochastic models adjusted to each position time series. We have extended this paragraph to specify this, and to better describe the figure.*

L335: Perhaps this sentence could be split into two for clarity.

*Yes, done.*

L347: should this be "but are mostly non-zero"?

*Yes, done.*

L378: should strict be strictly?

*Yes, "strict" now changed to "strictly".*

L388: Was there also improvement seen in the North & East components? If so, by how much?

*Yes, the figure below shows two panels that supplement Figure 8 in the manuscript. It shows improved results in the horizontal components too. However, we decided not to show these components to focus on the vertical component only, as the scope of the data paper is the vertical. If the reviewer or the editor insists, we can add the two panels below in the paper.*

[Figure]

L401: product's?

*"products" changed to "product".*

L402: suggest stating that it is the vertical velocity that experienced the reduction in uncertainty.

*"vertical' is added.*

Tables

Table 1
In the second column suggest writing out Earth Orientation Parameters tide model from Desai and Sibois (2016) as opposed to just the reference.

*Done.*

Figures

Fig. 1

Why are some station circles different sizes? If size as well as color corresponds to station duration, a key would be useful. Why does the record length range from 3 months to 21 years, wasn't there a selection cutoff of >3years? Are these shorter duration stations all French GNSS stations and/or reference frame stations?

*The issue of the circle size is fixed: now all circles have the same size. Yes, the stations shorter than 3 yrs correspond to some French stations.*

It may be useful to have subpanels with regions of higher concentration of stations, e.g. Europe, Western North America, Eastern North America. Or showing the regional subnetworks mentioned in the main text.

*The subpanels of Europe and North America have been added to Figure 1. The figure S1 showing the subnetworks distribution for the day 2018-01-01 has been added to the supplemental material.*

Fig. 2

Please label the x axis.
Suggest having the label for all be "ALL GPS" or "ALL GNSS" to make it clear that the GLOSS tide gauges are not included in the tally.

*"Date" is now added as x label. "ALL" is changed to "ALL GPS" and "GLOSS" changed to "GPS@GLOSS" for clarification*

Fig. 3

Could you increase the text size for the piechart labels? Perhaps change the color corresponding to the unknown category from red to purple to increase the color contrast for colorblind readers. Would also be useful to have a title for the piechart (e.g., "Offsets origin") to avoid needing to reference the caption.

*Thanks for these suggestions that improve the clarity of the Figure. The font size has been increased, the red has been changed to white, and black edge colour has been set for more contrast. A title has also been added.*

Fig. 4

Panel a -
Are there values in excess of 3 mm/yr? If so, please add triangles to the colorbar to indicate saturation at +/- 3mm/yr.
The stations with vertical velocities near 0 mm/yr are difficult to see. Stations could be outlined in black, or the colormap could be blue yellow red instead.

Panel b -
suggest switching colormaps to a sequential rather than diverging map. In particular, having the same red to blue colormap as in the above panel risks the reader thinking the panels are on the same scale.

*Thanks for these suggestions (adopted).*

Fig. 7

The colorbar for panel a might be better placed below the figure to avoid mistaking "record length [yr]" for the title of the panel.

*Changed, thank you.*

Fig. 8

It might be easier to read the histogram if the bar graph is filled in with transparent colors.

*Changed. Looks indeed better to some of us.*